Molecular mechanisms linking peri-implantitis and type 2 diabetes mellitus revealed by transcriptomic analysis

Yu Tianliang 1
http://orcid.org/0000-0002-9319-6780 Acharya Aneesha 2 3
http://orcid.org/0000-0001-7358-7496 Mattheos Nikos 2
Li Simin 4
http://orcid.org/0000-0002-9810-2368 Ziebolz Dirk 4
Schmalz Gerhard 4
Haak Rainer 4
Schmidt Jana 4
Sun Yu 1 sunyu20060212@126.com
1 Department of Prosthodontics, School of Dentistry, Harbin Medical University , Harbin, Heilongjiang , China
2 Faculty of Dentistry, University of Hong Kong , Hong Kong , China
3 Dr D Y Patil Dental College and Hospital , Pimpri, Pune , India
4 Department of Cariology, Endodontology and Periodontology, University Leipzig , Leipzig, Saxon , Germany
Yu Cheng-Chia
Electronic publication date: 2019 Jun 21
Publication date: 2019
Volume: 7
Electronic Location ID: e7124
Received 2019 Feb 18; Accepted 2019 May 14
Copyright: © 2019 Yu et al.
Copyright year: 2019
Copyright holder: Yu et al.
License: This is an open access article distributed under the terms of the Creative Commons Attribution License, which permits unrestricted use, distribution, reproduction and adaptation in any medium and for any purpose provided that it is properly attributed. For attribution, the original author(s), title, publication source (PeerJ) and either DOI or URL of the article must be cited.
License URL: https://creativecommons.org/licenses/by/4.0/

Keywords: Type 2 diabetes, Peri-implantitis, Gene, Pathway, Bioinformatics

Funding: China Scholarship Council (CSC) for Simin Li at University Leipzig 201608080010 The authors received doctoral study support from the China Scholarship Council (CSC) for Simin Li (CSC No: 201608080010) at University Leipzig. The funders had no role in study design, data collection and analysis, decision to publish, or preparation of the manuscript.

==============================
Aims

To explore molecular mechanisms that link peri-implantitis and type 2 diabetes mellitus (T2DM) by bioinformatic analysis of publicly available experimental transcriptomic data.

Materials and methods

Gene expression data from peri-implantitis were downloaded from the Gene Expression Omnibus database, integrated and differentially expressed genes (DEGs) in peri-implantitis were identified. Next, experimentally validated and computationally predicted genes related to T2DM were downloaded from the DisGeNET database. Protein–protein interaction network (PPI) pairs of DEGs related to peri-implantitis and T2DM related genes were constructed, “hub” genes and overlapping DEG were determined. Functional enrichment analysis was used to identify significant shared biological processes and signaling pathways. The PPI networks were subjected to cluster and specific class analysis for identifying “leader” genes. Module network analysis of the merged PPI network identified common or cross-talk genes connecting the two networks.

Results

A total of 92 DEGs overlapped between peri-implantitis and T2DM datasets. Three hub genes (IL-6, NFKB1, and PIK3CG) had the highest degree in PPI networks of both peri-implantitis and T2DM. Three leader genes (PSMD10, SOS1, WASF3), eight cross-talk genes (PSMD10, PSMD6, EIF2S1, GSTP1, DNAJC3, SEC61A1, MAPT, and NME1), and one signaling pathway (IL-17 signaling) emerged as peri-implantitis and T2DM linkage mechanisms.

Conclusions

Exploration of available transcriptomic datasets revealed IL-6, NFKB1, and PIK3CG expression along with the IL-17 signaling pathway as top candidate molecular linkage mechanisms between peri-implantitis and T2DM.

Introduction

With the increasing use of dental implants in oral rehabilitation, peri-implant inflammatory diseases have risen in incidence and comprise a significant clinical challenge. Peri-implant disease constitutes of peri-implant mucositis, characterized by reversible inflammation restricted to soft tissues around an implant and peri-implantitis, characterized by peri-implant alveolar bone loss due to progression of the inflammatory lesion (Renvert et al., 2018). Being multi-factorial, peri-implantitis is associated with a number of risk factors (Dreyer et al., 2018). Diabetes mellitus, a common metabolic disorder, is characterized by hyperglycemia resulting from insulin resistance, inadequate insulin secretion, or excessive glucagon secretion (Blair, 2016). Type 2 diabetes (T2DM), the main form of diabetes affecting 90–95% of diabetics, is a risk factor for peri-implantitis (Dreyer et al., 2018; Monje, Catena & Borgnakke, 2017), with a 50% higher risk reported in a meta-analysis (Monje, Catena & Borgnakke, 2017). Others have noted disease severity of peri-implantitis increases with worse glycemic control in T2DM (Al Amri et al., 2016; Gómez-Moreno et al., 2015).

Pathogenic mechanisms underlying T2DM mediated aggravation of peri-implant diseases are not as well-investigated and remain poorly understood. A common understanding is that impairment of vascularization and angiogenesis inherent to a T2DM microenvironment impairs bone healing by delaying wound healing, reducing bone formation, and impairing osteogenesis (Marin et al., 2018). Notably, diabetes mellitus alters host-immune responses, skewing these toward pro-inflammatory dominance (Nielsen et al., 2017). As a result, in diabetics, the cytokine production in response to peri-implant biofilm can be altered, when compared to that in healthy subjects. Among individuals with peri-implantitis, those having diabetes show an overproduction of multiple peri-implant fluid pro-inflammatory cytokines; IL-1, IL-6, IL-8, TNF-alpha, chemokine receptors CCR5 and CXR3 (Venza et al., 2010; Al-Askar et al., 2018), where the severity of dysregulation appears to be worsen with poor diabetes control. Such unfavorable immune modulation is likely to cause the higher susceptibility to peri-implantitis noted in diabetics (Monje, Catena & Borgnakke, 2017). Other mechanisms whereby peri-implantitis is aggravated in a poorly controlled diabetic mileu include potential alterations in biofilm composition (Gulia et al., 2018) and accumulation of advanced glycation end products (Al-Sowygh et al., 2018; Alrabiah et al., 2018). Nevertheless, to the authors’ knowledge, not much is known about the scale of molecular relationships between both diseases. The identification of linkage molecular mechanisms could translate to potential therapeutic targets for individualized treatment of peri-implantitis in T2DM affected subjects.

Integrated analysis of bioinformatic data has several advantages in scoping poorly understood disease associations. Relevant genomic data from multiple microarray and next-generation sequencing studies of single diseases can be used to determine molecular linkages through bioinformatic analyses. Several microarray or sequencing studies have examined global gene-expression patterns relevant to peri-implantitis (Becker et al., 2014; Schminke et al., 2015) and to T2DM (Berisha et al., 2011; Pihlajamäki et al., 2009; Taneera et al., 2013). An overview of these studies suggests some similarities in the molecular mechanisms implicated in both diseases. These include inflammatory cytokines (Interleukin-1β, Interleukin-6), vascular endothelial growth factor, and reactive oxygen species-related genes. By integrative bioinformatics, genomic data from T2DM and peri-implantitis documented in previous studies can facilitate the identification of potential molecular links. Therefore, a bioinformatic study of existing experimental transcriptome datasets was designed to investigate putative molecular links between peri-implantitis and T2DM by identifying cross-talk genes, biological processes (BPs), and signaling pathways involved in both diseases.

Materials and Methods

Procurement of data

Two transcriptome datasets pertaining to peri-implantitis (GSE33774 and GSE57631) were downloaded from the Gene Expression Omnibus database (http://www.ncbi.nlm.nih.gov/projects/geo/). The diagnostic criteria of peri-implantitis in each study was aligned with the current consensus case-definition of peri-implantitis: radiographic evidence of bone loss ≥3 mm and/or probing depths ≥6 mm in conjunction with profuse bleeding (Renvert et al., 2018). For the dataset GSE33774, a probing depth ≥5 mm in combination with radiographic bone loss around implants >3 mm were indicative of peri-implantitis (Becker et al., 2014). For the dataset GSE57631, the diagnostic criteria of peri-implantitis were not specified in the relevant text (Schminke et al., 2015) and by contacting the authors (Schminke et al., 2015), these were determined to be consistent with the current consensus criteria (Renvert et al., 2018). Experimentally validated and computationally predicted genes associated with T2DM were downloaded from the DisGeNET database (http://www.disgenet.org/home/). The DisGeNET database is a comprehensive platform that integrates information concerning human disease-associated genes and their variants (Piñero et al., 2016). This database integrates data from expert curated repositories, text mining data extracted from scientific literature, experimentally validated data, and referred data (Piñero et al., 2016). By using this database, a total of 1,274 T2DM-associated genes were determined.

Differential gene expression and functional enrichment analysis

The two peri-implantitis-related transcriptome datasets were subjected to differential expression analysis using the “limma” package in R (Ritchie et al., 2015). Peri-implantitis-related genes with p-value < 0.05 and |logFC| ≥ 1 were screened and defined as differentially expressed genes (DEGs). Next, enrichment analysis was performed with the R package “clusterProfiler” to describe the Gene Ontology (GO) and Kyoto Encyclopedia of Genes and Genomes (KEGG) functional profiles of the DEGs related to peri-implantitis.

Protein–protein interaction network analysis

Protein–protein interaction (PPI) pairs of DEGs related to peri-implantitis and genes related to T2DM were each determined using the free web-available database Search Tool for the Retrieval of Interacting Genes (Release 9.1, http://string-db.org/), as it has medium to high confidence scores. Based on these PPI pairs, PPI networks were constructed for peri-implantitis and T2DM, respectively, and the topological characteristics of the two PPI networks were analyzed.

Clustering analysis

A “combined association score” (CAS) of each PPI pair, representing the interacting pair’s degree of confidence, was determined. The CAS for each gene over that of its neighbors was summed and defined as “comscore,” which represented the number of weighted links for that gene. K-means clustering analysis was applied to these “comscores” and gene nodes were classified into five classes according to their weighted number of links. Specific class analysis was performed by determination of K-means based classes specific to and significantly associated with peri-implantitis and T2DM each, by using analysis of variance and multiple-testing with Tukey–Kramer test (at p < 0.01). Genes in these classes were further analyzed. Genes belonging to the highest rank were defined as “leader genes,” as these genes had the highest weighted number of links as compared to others in that network. The “leader gene” approach enables the identification of a small number of potentially most relevant candidate genes from the experimental dataset.

Cross-talk gene analysis

Protein–protein interaction networks for peri-implantitis and T2DM were merged to construct a global network. T2DM related genes among the DEGs associated to peri-implantitis were identified as cross-talk genes. Module networks of these cross-talk genes were constructed using a graph theoretic clustering algorithm “Molecular Complex Detection.” Modules represent genes that are densely connected in a PPI network are likely to represent highly related protein complexes or modules.

Results

Identification of DEGs

The DEGs identified in peri-implantitis are listed in Table S1. A total of 224 DEGs were identified in GSE33774 and 813 DEGs were identified in GSE57631. Combining these, a total of 1,028 genes emerged as associated with peri-implantitis. For T2DM, a total of 1,274 genes were obtained. A total of 92 of the DEGs were shared by peri-implantitis and T2DM (Fig. S1).

Functional enrichment analysis

Differentially expressed genes in peri-implantitis were mainly involved in the BPs of neutrophil activation, neutrophil mediated immunity, antigen processing and presentation of peptide antigen via MHC class I, and positive regulation of protein catabolic process (Fig. S2). Genes associated with T2DM were mainly enriched in BPs of response to nutrient levels and response to oxidative stress (Fig. S3). No BP emerged as overlapped between peri-implantitis and T2DM. In terms of KEGG signaling pathways, DEGs associated with peri-implantitis were seen as mainly related to protein processing in endoplasmic reticulum (ER), IL-17 signaling pathway, and Leukocyte transendothelial migration (Fig. S4). Genes related to T2DM were significantly enriched in pathways related to insulin resistance and FoxO signaling (Fig. S5). Combining these, a single signaling pathway; IL-17 signaling, was seen as shared by peri-implantitis and T2DM.

PPI network analysis

The PPI network in peri-implantitis showed 786 nodes and 5,278 interaction pairs (Fig. 1A) whereas, that in T2DM showed 1,171 nodes and 26,862 interaction pairs (Fig. 1B). The topological characteristics of the two PPI networks are summarized in Table S2, where gene nodes are ranked in descending order of their degree, and the top 20 nodes or “hub genes” were determined. Among these top 20 hub genes, three genes (IL6, NFKB1, and PIK3CG) were common to peri-implantitis and T2DM, thus can be regarded as the potential cross-talk genes. Considering these hub genes appeared to be central to the regulation of and may prominently impact the biological network, PPI subnetworks of these three genes were extracted (Fig. 2). In the leader gene approach, cluster analyses for peri-implantitis showed class 2 as the most significantly associated class (Fig. S6A) and for T2DM, class 3 was the most significant (Fig. S6B). Extracting the genes in these classes, three leader genes (PSMD10, SOS1, and WASF3) were found to be overlapping (Table S3).

Figure 1 The PPI network of DEGs expressed in peri-implantitis (A) and PPI network of genes related to type 2 diabetes (B).

Figure 2 The PPI subnetwork of IL6 (A), NFKB1 (B), and PIK3CG (C).

Module network analysis

Module network analysis of the cross-talk genes showed three subnetworks. Module subnetwork 1, where red nodes represent the common genes, green nodes represent T2DM genes and blue nodes are DEGs associated with peri-implantitis, showed PSMD10 and PSMD6 as cross-talk genes, which directly interacted with one gene closely related to diabetes (LTA), as well as with DEGs in peri-implantitis (PSM family genes (PSMC2, PSMA4, PSME4, etc.), SKP2, EIF3F, USP14, UBLCP1, etc.) (Fig. 3). In module subnetwork 2 (Fig. 4), five cross-talk genes; EIF2S1, GSTP1, DNAJC3, SEC61A1, and MAPT were noted, which directly interacted with a few T2DM genes and DEGs of peri-implantitis. Module subnetwork 3 (Fig. 5), showed only one common gene (NME1), interacting with small number of genes involved in both networks.

Figure 3 The module network 1 identified two cross-talk genes (PSMD1 and PSMD6).

Figure 4 The module network 2 identified five cross-talk genes (EIF2S1, GSTP1, DNAJC3, SEC61A1, and MAPT).

Figure 5 The module subnetwork 3 identified only one cross-talk gene (NME1).

Known functions of the key hub genes and cross talk genes in T2DM and peri-implantitis each are described in Table 1.

Table 1 The functions of genes identified in the pathogenesis of T2DM and peri-implantitis, respectively.

Genes	General functions	Functions in T2DM	Functions in peri-implantitis	
IL-6	Interleukin 6 (IL-6) is an interleukin that acts as both a pro-inflammatory cytokine and an anti-inflammatory myokine	Has an anti-inflammatory role and improves glucose metabolism;

Induces the development of insulin resistance and pathogenesis of T2DM through the generation of inflammation by controlling differentiation, migration, proliferation, and cell apoptosis

	Exert pro-inflammatory effects;

Induce bone resorption

	
NFKB1	NFKB1 (Nuclear Factor Kappa B Subunit 1) encodes a 105 kD protein which is a DNA binding subunit of the NF-kappa-B (NFKB) protein complex	Be involved in a compensatory mechanism that develops in β-cells during the loss of insulin sensitivity;

Be implicated in the expression of GLUT2, which contributes to glucose-stimulated insulin secretion by β-cells;

Its inhibition may have deleterious effects leading to the development of insulin resistance and type 2 diabetes

	Regulate the inflammatory-induced osteoclastogenesis process;

Regulate receptor activator of NF-κB ligand (RANKL)—mediated osteoclast formation and activation

	
PIK3CG (also called PI3K)	A family of lipid kinases that catalyze the phosphorylation of plasma membrane lipid phosphatidylinositol;

Being involved in PI3K/AKT pathway

	Its reduction impairs insulin signal transduction, resulting in the impaired translocation of glucose transporter protein GLUT4 and insulin resistance	Be correlated with inflammation regulation, angiogenesis, and osteoclast activity;

PI3K signaling can lead to osteogenic induction and increased osteogenic differentiation of Periodontal Ligament Stem Cells (PDLSCs), thus PI3K could be involved in peri-implantitis via regulation of peri-implant osteogenesis

	
SOS1	SOS1 (SOS Ras/Rac guanine nucleotide exchange factor 1) regulates Ras (Rat sarcoma) proteins by aiding the exchange of GTP for GDP	Regulates Ras proteins, which implicates in the development of diabetic vascular dysfunction by inducing abnormal vascular reactivity	Regulates Ras proteins, which is an early signal for osteogenesis	
WASF3	WASF3 (Wiskott–Aldrich syndrome protein family member 3) plays a role in the regulation of cell morphology and cytoskeletal organization	Plays a role in the remodeling of actin cytoskeleton, which is involved in the regulation of pancreatic β-cell insulin secretion.

Regulate cell morphology; red cell morphology changes have been observed in T2DM patients with high prevalence

	Actin binding was shown to be significant biological process in peri-implantitis, thus WASF3 may be involved in peri-implantitis by controlling actin binding	
PSMD10	PSMD10 (Proteasome 26S Subunit, Non-ATPase 10), PSMD6 (Proteasome 26S Subunit, Non-ATPase 6) is regulatory component for the 26S proteasome, which is central to protein regulation by ubiquitination-degradation	Altered ubiquitin-proteasome system (UPS) might be one of the molecular mechanisms of insulin resistance in T2DM;

Altered activity of UPS can contribute to the development of retinal microvascular complications of diabetes

	The activation of the ubiquitin–proteasome is involved in an NF-κB-dependent increase in peri-implant inflammation;

DEGs expressed in peri-implantitis were examined to be enriched in proteasomal ubiquitin-dependent protein catabolic process

	
EIF2S1	EIF2S1 (Eukaryotic translation initiation factor 2 subunit 1) is a component of the PI3K pathway, encoding for eukaryotic initiation factor 2α(eIF2α), the phosphorylation of which can reduce protein synthesis	Dysregulation of eIF2α phosphorylation is poorly tolerated by pancreatic β cells, leading to dysfunction	Downregulates infection-induced cytokine expression, thus involved in the immune inflammatory response	
GSTP1	GSTP1 (Glutathione S-Transferase Pi 1) is strongly associated with the metabolic efficiency, detoxification, and inflammatory diseases and cancer susceptibility	Decrease ROS species and act as a kind of antioxidant defense;

Since oxidative stress is involved in T2DM, thus the association of GSTP1 and T2DM has been investigated by many studies. However, with controversial results

	The presence of GSTP1 polymorphism may be a risk factor for the development of chronic periodontitis;

The association of GSTP1 and peri-implantitis hasn’t yet been investigated

	
DNAJC3	P58IPK (DNAJ Heat Shock Protein Family (Hsp40) Member C3, also called P58IPK) functions as a signal for the downregulation of endoplasmic reticulum (ER)-associated proteins involved in the initial ER stress response	Be involved in the development of T2DM since the disruption of its mediated ER can cause the dysfunction of insulin-secreted beta cells	Be involved in peri-implantitis by activating the unfolded protein response (UPR) pathway associated with inflammation and alveolar bone resorption	
SEC61A1	SEC61A1 (Sec61 Translocon Alpha 1 Subunit) plays a crucial role in the insertion of secretory and membrane polypeptides into the endoplasmic reticulum (ER)	Has the similar mechanism with DNAJC3: be involved in the development of T2DM since the disruption of its mediated ER can cause the dysfunction of insulin-secreted beta cells	Has the similar mechanism with DNAJC3: be involved in peri-implantitis by activating the unfolded protein response (UPR) pathway associated with inflammation and alveolar bone resorption	
MAPT	MAPT (microtubule associated protein Tau), a neural phosphoprotein member of the MAP family, is implicated in microtubule function within the cell cytoskeleton	Disturbance in MAPT phosphorylation is shown to decrease insulin production from pancreatic beta cells	Be implicated in bone mineral density regulation	
NME1	NME1 (NME/NM23 Nucleoside Diphosphate Kinase 1) is a negative regulator of nuclear factor-κB (NF-kB) signaling which is a critical player in immune responses	The inhibited activity of NF-kB can improve conductance artery function in T2DM, thus NME1 may be involved in the T2DM by impairing the artery function	Inhibition of NF-κB can prevent the inflammatory response during peri-implantitis, thus NME1 may be involved in peri-implantitis;

Receptor activator of nuclear factor kappa B (RANK) was shown to be a pathologic determinant of peri-implantitis, thus NME1 may be involved in peri-implantitis

	

Discussion

The current study included both peri-implant bone and peri-implant soft tissue transcriptomes and the findings can be considered as an integrated or non-specific view of peri-implantitis-T2DM molecular linkages. Three common or cross-talk genes; IL-6, NFKB1, and PIK3CG, were noted among the top 20 hub genes in the PPI networks of T2DM and peri-implantitis. Raised interleukin (IL)-6 has been demonstrated as an independent predictor of T2DM (Akbari & Hassan-Zadeh, 2018). IL-6 promotes inflammation and can induce insulin resistance (Rehman et al., 2017). IL-6 expression is also increased in peri-implantitis, reflecting its role in destruction of peri-implant tissue and bone resorption by exerting pro-inflammatory effects (Candel-Martí et al., 2011). Raised IL-6 in the peri-implant crevicular fluid has been proposed as a marker of peri-implantitis (Yaghobee et al., 2014). The NFKB1 gene is a transcription factor which encodes the nuclear-factor kappa beta (NF-kB) p105/p50 isoforms (Héron, Deloukas & Van Loon, 1995). NF-kB inflammatory signaling is implicated in the development of T2DM (Baker, Hayden & Ghosh, 2011) and experimental evidence shows interfering with NF-kB signaling can decrease hyperglycemia and insulin resistance. In peri-implantitis, NFKB1 has been previously identified as a key candidate gene (Zhang et al., 2017) and is shown to regulate inflammation-induced osteoclastogenesis by regulating receptor activator of NF-κB ligand (RANKL)—mediated osteoclast formation and activation in peri-implantitis (Boyce et al., 2015). The phosphoinositide-3 kinase (PI3K, also called PIK3CG) is an important regulator of cell response to extra-cellular stimuli. PI3KCG gene encodes a class I catalytic subunit of PI3K protein, which can bind a p85 regulatory subunit to form PI3K (Amzel et al., 2008). In T2DM, PI3K activity reduction impairs insulin signal transduction and impairs translocation of glucose transporter protein GLUT4 leading to insulin resistance (Niswender et al., 2003). PI3K signaling is also shown to increase the osteogenic differentiation of periodontal ligament stem cells (Lee et al., 2014a), thus, could be implicated in peri-implantitis via regulation of osteogenesis. Thus, experimental evidence supports the notion that the in silico determined shared genes could be significant mechanistic links between T2DM and peri-implantitis.

Three common leader genes; PSMD10, SOS1, and WASF3, were identified from the gene clusters linked to peri-implantitis and T2DM each. Proteasomes are involved in intracellular protein degradation and implicated in several diseases. The PSMD10 or Gankyrin gene (proteasome 26S subunit, non-ATPase 10) encodes subunits of the 26S proteasome, a component of the ubiquitin–proteasome (UPS) system (Hochstrasser, 1996). It has been found enriched in inflammation (Lecker, Goldberg & Mitch, 2006) and may be induced by pro-inflammatory Interleukin-1 beta stimulation (Qureshi, Morrison & Reis, 2012). It is upregulated in T2DM (Costes et al., 2011), and was also implicated previously in peri-implantitis (Zhang et al., 2017) but no experimental evidence exists. SOS1 (SOS Ras/Rac guanine nucleotide exchange factor 1) regulates Rat sarcoma proteins and facilitates exchange of GTP for GDP. In T2DM, Ras-GTPase has been implicated in inducing aberrant vascular reactivity and dysfunction (Yousif et al., 2004). In peri-implantitis, while SOS1 was previously identified as a hub gene (Zhang et al., 2017) but experimental evidence is similarly lacking. Ras activation is an early signal for osteogenesis in human bone marrow stromal cells (Wang et al., 2001). The Ras superfamily is also suggested to regulate inflammation by acting as regulators of NF-κB and Ral pathways (Oeckinghaus et al., 2014). WASF3 (WAS protein family member 3) controls actin binding, thus regulating cell shape and motility, and has been frequently implicated in cancer cell motility and metastasis (Teng et al., 2016). As glucose transporter recruitment is actin dependent, a plausible role of WASF3 signaling in T2DM pathology is suggestible (Tunduguru et al., 2017). In addition, actin binding was found as a significantly dysregulated process by a whole-exome sequencing study of peri-implantitis (Lee et al., 2014b), but no experimental study has yet characterized WASF3’s expression in peri-implantitis.

Module network analysis identified eight cross-talk genes; PSMD10, PSMD6, EIF2S1, DNAJC3, SEC61A1, GSTP1, MAPT, and NME1. Like PSMD10, PSMD6 is regulatory for the 26S proteasome, which is central to protein regulation by ubiquitination-degradation. UPS dysregulation might be a molecular mechanism underlying insulin resistance (Balasubramanyam, Sampathkumar & Mohan, 2005) and has also been implicated in microvascular complications of T2DM (Aghdam & Sheibani, 2013). In peri-implantitis, proteasomal ubiquitin-dependent protein catabolic process (GO term: 0043161) was found to be significantly enriched (Zhang et al., 2017). Based on these perspectives, it may be hypothesized that UPS dysregulation in a T2DM state contributes to aggravated peri-implantitis via a positive feedback loop, wherein pro-inflammatory cytokine responses to peri-implant biofilm may further its dysregulation. EIF2S1 is a component of the PI3K pathway, encoding for the eukaryotic initiation factor 2α (eIF2α) involved in regulating protein synthesis (Jiang & Wek, 2005) and ER stress responses. Activation of eIF2α-mediated signaling by bacterial pathogens downregulates infection-induced cytokine expression (Shrestha et al., 2012) and may propagate the inflammatory processes. It is also implicated in T2DM, as dysregulation of eIF2α leads to pancreatic β cells dysfunction (Cnop et al., 2017). The expression pattern or effects of eIF2α in peri-implantitis are not specifically investigated. Among the genes noted in module subnetwork 2, DNAJC3 (DnaJ Heat Shock Protein Family (Hsp40) Member C3 or P58IPK) inhibits eIF-2α signaling, thereby attenuating the later phases of the ER stress response (Ladiges et al., 2005). DNAJC3 mutation is implicated in the development of T2DM via dysfunction of insulin-secreting beta cells (Ladiges et al., 2005). In peri-implantitis it can activate the unfolded protein response pathway associated with inflammation and alveolar bone resorption (Yamada et al., 2015). The possible mechanisms of SEC61A1 (Sec61 Translocon Alpha 1 Subunit) in T2DM and peri-implantitis appear to be similar to those of the DNAJC3 gene. SEC61A1 is also related to ER stress response by control of polypeptide transport into ER (Lang et al., 2012). Glutathione S-Transferase Pi 1 (GSTP1) is suggested to protect cells against oxidative stress (Savic-Radojevic et al., 2007), which is involved in both peri-implantitis (Sánchez-Siles et al., 2016) and T2DM (Wright, Scism-Bacon & Glass, 2006) pathology. GSTP1 polymorphism is associated with susceptibility to type II diabetes mellitus (Saadat, 2017) and the risk for developing chronic periodontitis (Camargo Ortega et al., 2014) and could similarly imply risk for its analogue peri-implantitis (Dhir et al., 2013). The microtubule associated protein Tau (MAPT), a member of the MAP family regulated microtubule function within the cell cytoskeleton (Sündermann, Fernandez & Morgan, 2016). Disturbance in MAPT phosphorylation is shown to decrease insulin production from pancreatic beta cells, supporting its role in T2DM pathology (Maj et al., 2016). MAPT gene polymorphism is also implicated in bone mineral density regulation (Dengler-Crish, Smith & Wilson, 2017). NME1 (NME/NM23 Nucleoside Diphosphate Kinase 1) is a known negative regulator of nuclear factor-κB signaling (You et al., 2014). Considering nuclear factor-κB signaling has been shown to be dysregulated and involved in pathogenesis of both T2DM (Andreasen et al., 2011) and peri-implantitis (Rakić et al., 2013), NME1 may be a relevant upstream molecular link between T2DM and peri-implantitis.

IL-17 signaling emerged as the sole pathway significantly shared between T2DM and peri-implantitis. IL-17 plays critical regulatory roles in host defense and inflammatory diseases showing both protective and destructive effects (Zou et al., 2013). Although IL-17 protects against pathogen invasion at epithelial or mucosal barriers, its dysregulation can stimulate overexpression of pro-inflammatory cytokines (IL-1β, IL-6, TNF-α) leading to continued tissue damage (Jin & Dong, 2013). IL-17 overexpression is noted in peri-implant crevicular fluid of peri-implantitis (Severino et al., 2016) and serum of T2DM (Nadeem et al., 2013). IL-17 negatively impacts osteogenesis in peri-implantitis affected tissue (Kim et al., 2014). In T2MD, IL-17 can contribute to the exacerbation of insulin resistance by inducing apoptosis of pancreatic β-cells (Yousefidaredor et al., 2014). Considering that IL-17 antagonist molecules have recently emerged as therapeutics, this finding may suggest a basis to explore their therapeutic potential in T2DM affected peri-implantitis patients (Abdel-Moneim, Bakery & Allam, 2018).

Taken together, perspectives from prior literature support the biological basis for many of the significant genes and pathways that emerged as putative mechanistic links between peri-implantitis and T2DM. The major limitation of this study is the lack of experimental validation of the genes highlighted in the in silico analyses which was beyond the scope of the current investigation. Thus, these findings have significant implications for future research. Most of the linkage genes revealed by the study lack experimental evidence in context of peri-implantitis-T2DM disease association. In theory, these molecular entities could be valuable as potential targets for individualized gene therapy, risk stratification, and therapy of peri-implantitis in type 2 diabetes. Further validation of key molecular mechanisms could promote the development of targeted drugs for blocking or aiding their expression and modulating related pathways. Most importantly, the present findings may be considered as hypotheses for future validation experiments and direction for research. As such, bioinformatic data mining of experimental transcriptomes is exploratory and at best considered as a source of well-supported hypotheses. Validation experiments could include comparison of the highlighted genes’ expression levels in the peripheral blood and peri-implant crevicular fluid of peri-implantitis affected individuals with and without T2DM and explore their roles in vitro/animal disease models.

Conclusion

Bioinformatics analysis combining experimental transcriptomic data from T2DM and peri-implantitis revealed potentially shared molecular linkages. Three hub genes (IL-6, NFKB1, and PIK3CG) identified in PPI networks, three cross-talk genes (PSMD10, SOS1, and WASF3) identified by specific class analysis and eight cross-talk genes (PSMD10, PSMD6, EIF2S1, GSTP1, DNAJC3, SEC61A1, MAPT, and NME1) obtained by module network analysis, and IL-17 signaling emerged as top candidate shared molecular linkages. Future studies should explore their roles in context of T2DM-peri-implantitis disease association.

Supplemental Information

Supplemental Information 1 The overlapping 92 genes between DEGs of peri-implantitis and experimentally validated genes of T2DM.

Click here for additional data file.

Supplemental Information 2 The top 30 biological process (BP) enriched in DEGs associated with peri-implantitis.

Click here for additional data file.

Supplemental Information 3 The top 30 BP enriched in genes related to type 2 diabetes mellitus.

Click here for additional data file.

Supplemental Information 4 All the pathways enriched in DEGs related to peri-implantitis.

Click here for additional data file.

Supplemental Information 5 All the pathways enriched in genes related to type 2 diabetes mellitus.

Click here for additional data file.

Supplemental Information 6 The comparison between different classes in peri-implantitis (a) and type II diabetes (b). The horizontal axis represents the different class, and the vertical axis represents the comscore corresponding to the class. The letter (a-e) shows the difference b.

Click here for additional data file.

Supplemental Information 7 Tables S1-S3.

Table S1. The number of DEGs identified in two datasets (GSE33774 and GSE57631) of peri-implantitis.

Table S2. Top 20 nodes in PPI networks of both peri-implantitis and T2MD.

Table S3. Three leader genes shared in two selected significant classes.

Click here for additional data file.

We are grateful to Ms. Xiangqiong Liu and Mr. Yupei Deng, who are bioinformatics engineers at Shanghai Genomap Technologies, Shanghai, China. We have to express our appreciation to them for providing us technological assistance during the analysis of this research.

Additional Information and Declarations

Competing Interests

Author Contributions

Data Availability

The authors declare no potential conflict of interests with respect to the authorship and publication of this paper.

Tianliang Yu analyzed the data, contributed reagents/materials/analysis tools, prepared figures and/or tables, authored or reviewed drafts of the paper, approved the final draft.

Aneesha Acharya prepared figures and/or tables, authored or reviewed drafts of the paper, approved the final draft, proof-reading.

Nikos Mattheos approved the final draft, proof-reading.

Simin Li analyzed the data, prepared figures and/or tables, authored or reviewed drafts of the paper, approved the final draft.

Dirk Ziebolz analyzed the data, approved the final draft, proof-reading.

Gerhard Schmalz analyzed the data, contributed reagents/materials/analysis tools, prepared figures and/or tables, approved the final draft.

Rainer Haak contributed reagents/materials/analysis tools, prepared figures and/or tables, approved the final draft.

Jana Schmidt contributed reagents/materials/analysis tools, approved the final draft.

Yu Sun authored or reviewed drafts of the paper, approved the final draft.

The following information was supplied regarding data availability:

No primary data was generated in this study. The publicly available microarray datasets (GSE33774 and GSE57631) were downloaded from Gene Expression Omnibus (GEO) database.

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
