# Peer review of "Molecular mechanisms linking peri-implantitis and type 2 diabetes mellitus revealed by transcriptomic analysis"

_PeerJ, doi:10.7717/peerj.7124_

## Round 0.1 · original submission · Minor Revisions

Please respond to the critiques from the reviewers.

Reviewer 1 ·

Basic reporting

-The language is professional and clear.
- The research question is clearly stated and no outlandish conclusions are drawn

Experimental design

- How many datasets were available in GEO for this study? What were the criteria for choosing these two datasets?

- Please rephrase line 25-26

- Please explain more clearly what 'experimentally validated genes from T2DM' means. Where was this differential expression observed? In blood, or tissue or some other particular sub-condition? It is important to know in what context these genes are associated with T2DM

Validity of the findings

Results

-line 132-133: What explains the 4-fold difference in DEGs between the two selected datasets?

- T2DM is the pre-existing condition, so discussion of results should be ordered with T2DM first then followed by peri-implantitis, which presents a clearer picture.

- Comparison of peri-implantitis in normal vs T2DM patients will give a more complete picture of the differential mechanisms

Reviewer 2 ·

Basic reporting

no comment

Experimental design

no comment

Validity of the findings

no comment

Additional comments

Yu et al in their manuscript entitled “Molecular mechanisms linking peri-implantitis and type II diabetes mellitus revealed by transcriptomic analysis” performed a bioinformatic study of existing experimental datasets, aimed to determine putative molecular links between peri-implantitis and T2DM by identifying cross-talk genes, biological processes and signaling pathways involved in both diseases.
The results concluded that 3 hub genes IL-6, NFKB1, and PIK3CG expression along with the IL-17 signaling pathway were the top candidate molecular linkage mechanisms between peri-implantitis and T2DM.

In general, as was mentioned a limitation, this study arose from bioinformatic data mining of experimental transcriptomes, and lack of experimental validation in clinical or in-vitro studies. Thus the results could only be regarded as hypotheses-driven data.

Major point:
A table is needed for briefly describing the functions of the 3 hub genes (IL-6, NFKB1, and PIK3CG), 3 cross-talk genes (PSMD10, SOS1, and WASF3) and 8 cross-talk genes (PSMD10, PSMD6, EIF2S1, GSTP1, DNAJC3, SEC61A1, MAPT, and NME1).

Minor points
1. Type 2 diabetes, instead of type II diabetes (title, line 104, and Fig 1 legend)
2. line 126, using a (redundant repeat)
3. line 193, “Three” common leader genes; instead of “3” common leader genes

·

Basic reporting

The review is very well written piece of scientific work. It is clear what the study wanted to achieve, how it was done and the results/discussion of the article is clear.

Experimental design

They used the most appropriate software available in this bioinformatic analysis. The research methods were well executed, rigorous using multiprong analysis from genes to the protein interaction to establish co-relation between the two disease. Also with the existence of only a handful of literature or studies that correlates T2DM with peri-implantitis, this study stands out to be a needed one to fulfill the present gap.

Validity of the findings

The results/discussion and conclusion section of the study were well described overall. Whenever applicable proper statistical methods were applied to discern the significance of the study. The study at the end lists the limitation and suggests potential in-vitro experiments needed to be conducted to verify the current findings.

Additional comments

Nice and scientifically stimulating work.

Reviewer 4 ·

Basic reporting

Basically, this article is clear and well-organized. However, the rationals to do this experiment to investigate the possible linking between T2DM and peri-implantitis are relative weak and vague. I hope the author should emphasize why this study was initiated and why?

Experimental design

The experimental design is fine. But the author should explain a little bit more about why the author choose this kind of software, such as network analysis and...etc. because current there is many software available.

Validity of the findings

The novelty is fine and this approaches should be an alternative method to explain the link between T2DM and peri-implantitis.

Additional comments

This rationale of this study, and clinical relevant of this study should be emphasized and re-organized.

---

## Round 0.2 · accepted · Accept

In my opinion the authors have satisfied the critiques from the reviewers